# Using lexical language models to detect borrowings in monolingual wordlists

**John E. Miller**[1☯*], **Tiago Tresoldi**[2☯*], **Roberto Zariquiey**[3], **César A. Beltrán Castañón**[1], **Natalia Morozova**[2], **Johann-Mattis List**[2]

**1** Artificial Intelligence/Engineering, Pontificia Universidad Católica del Perú, San Miguel, Lima, Peru, **2** Department of Linguistic and Cultural Evolution, Max Planck Institute for the Science of Human History, Jena, Germany, **3** Humanities Department, Pontificia Universidad Católica del Perú, San Miguel, Lima, Peru

☯ These authors contributed equally to this work.
\* jemiller@pucp.edu.pe (JEM); tresoldi@shh.mpg.de (TT)

**Data Availability Statement:** All data and code files are available from https://doi.org/10.5281/zenodo.4244667 as PyBor: A Python library for borrowing detection.

## Abstract

Lexical borrowing, the transfer of words from one language to another, is one of the most frequent processes in language evolution. In order to detect borrowings, linguists make use of various strategies, combining evidence from various sources. Despite the increasing popularity of computational approaches in comparative linguistics, automated approaches to lexical borrowing detection are still in their infancy, disregarding many aspects of the evidence that is routinely considered by human experts. One example for this kind of evidence are phonological and phonotactic clues that are especially useful for the detection of recent borrowings that have not yet been adapted to the structure of their recipient languages. In this study, we test how these clues can be exploited in automated frameworks for borrowing detection. By modeling phonology and phonotactics with the support of Support Vector Machines, Markov models, and recurrent neural networks, we propose a framework for the supervised detection of borrowings in mono-lingual wordlists. Based on a substantially revised dataset in which lexical borrowings have been thoroughly annotated for 41 different languages from different families, featuring a large typological diversity, we use these models to conduct a series of experiments to investigate their performance in mono-lingual borrowing detection. While the general results appear largely unsatisfying at a first glance, further tests show that the performance of our models improves with increasing amounts of attested borrowings and in those cases where most borrowings were introduced by one donor language alone. Our results show that phonological and phonotactic clues derived from monolingual language data alone are often not sufficient to detect borrowings when using them in isolation. Based on our detailed findings, however, we express hope that they could prove to be useful in integrated approaches that take multi-lingual information into account.

## Introduction

### Problem and motivation

Lexical borrowing (i.e., the direct transfer of words from one language to another) is one of the most frequent processes of language evolution [1]. We can easily observe the process in real time, especially regarding vocabulary from religion or technology, since words are often

**Funding:** JEM, has received funding and encouragement from the Graduate School of the Pontificia Universidad Católica del Perú (PUCP) through the Huiracocha-2019 scholarship program (https://posgrado.pucp.edu.pe). TT, JML, have received funding from the European Research Council (ERC) under the European Union's Horizon 2020 research and innovation programme (grant agreement No. ERC Grant #715618, "Computer-Assisted Language Comparison"). (https://erc. europa.eu). RZ, has received funding from Pontificia Universidad Católica del Perú (PUCP) through the project (604 DGI-PUCP) ¿Gramáticas que mueren?: Aproximación crítica a la obsolescencia de las lenguas desde la documentación y la tipología lingüísticas, las ciencias de la información y la inteligencia artificial. The funders had no role in study design, data collection and analysis, decision to publish, or preparation of the manuscript.

**Competing interests:** The authors have declared that no competing interests exist.

transferred along with cultural practices or innovations. While it took scientists a long time to find out that languages constantly change [2], it was already clear in ancient times that languages acquire lexical material from their neighbors [3], as evidenced in Plato's *Kratylos* dialog (409d-10a) [4] where Socrates discusses the problem that lexical borrowings impose on studies in etymology. Nonetheless, detecting borrowings is still one of the outstanding problems in historical linguistics, specifically when it comes to computational approaches [5].

Discrimination between inherited and borrowed words (or *loanwords*) is crucial for the successful application of both the *comparative method* in historical linguistics [2], which seeks to identify genetically related languages and reconstruct their ancestral stages which are not recorded in written sources, and in *phylogenetic reconstruction*, which seeks to identify the most plausible phylogenies (often represented by a family tree) by which languages in a given family evolved into their current shape [6].

Lexical borrowing is a very peculiar process that cannot be directly compared with other processes of language change. While sound change, for example, proceeds in a surprisingly regular manner that tends to impact all words in the lexicon of a particular spoken language, in which the sound occurs in a given phonetic environment, it is obvious that lexical borrowing crucially depends on the initial *contact situation*. This means also that language contact will impact differently on particular languages, depending on their geographic location or the interaction with speakers from other cultures of their original speakers. For this reason, it has also proven very difficult to come up with general statistics on lexical borrowing, and although scholars seem to agree in general that some words are less likely to be borrowed, depending on the meanings they express [7, 8], it is extremely difficult to derive generalizations, given that individual language histories provide many surprises [9].

When linguists try to detect borrowings, they use an arsenal of different techniques which essentially aim at detecting *conflicts* in the data for individual words [10]. English *mountain* for example, which was borrowed from Old French, exhibits a *phylogenetic conflict*, given its similarity to the words denoting 'mountain' in the Romance languages (cf. Italian *monte*, Spanish *montaña*) as opposed to its lack of similarity with words in Germanic languages (cf. German *Berg*, Dutch *berg*). Since English is a Germanic language, the word *mountain* shows a phylogenetic conflict, given its similarity with words from Romance languages. As another kind of conflict, German *Damm* 'dam' is obviously similar to its translational equivalent *dam* in English, but the similarity is not expected, since English words starting in *d* tend to have related words starting in *t* in German, and we observe a conflict with established *sound correspondences*.

While computational approaches to borrowing detection have tried to detect at least some of these conflicts systematically [10], there is another type of conflict that has not yet been explored so far. This conflict is reflected in the fact that at least in some languages and under specific circumstances borrowed words may still retain certain foreign properties when they have just entered a given language. These characteristics include both specific phonological properties ('foreign sounds') as well as specific *phonotactic* properties ('foreign distributions of sounds'), and they tend to disappear with time, due to the process of *loanword nativization* [11, p. 200].

Although masked with time, *language-internal evidence for borrowing* can be observed in many languages from different families. In many Hmong-Mien languages, for example, some Chinese words are borrowed with a very specific tone that only occurs in Chinese words [12]. Similarly, it is easy for German speakers to identify *job* as a loan from English, since only in borrowed words the grapheme *j* is pronounced as [dZ] in German. In the same line, but in a radically different context, speakers of Iskonawa, an obsolescent Panoan language spoken in Central Peruvian Amazonia can easily identify loanwords from Shipibo-Konibo, the dominant

language in the area, due to straightforward phonological features. For instance, Iskonawa has dropped word-initial [h], thus forms like [hana] 'tongue' or [huni] 'man' are easily detected as loanwords from Shipibo-Konibo.

Apart from specific sounds and tones, language-internal evidence for borrowing may include peculiar constructions, specific phonotactic elements (such as certain consonant clusters or vowel combinations), unusual stress patterns [13, 14], or even specific semantics or morphology. For instance, in Spanish words like *análisis* 'analysis' or *curriculum* 'curriculum' are easily identified as loans due to their irregular plural forms, *análisis* (invariant) and *curricula* (with a final *a*), respectively.

However, already upon entering the language, speakers adapt borrowed words to the phonological conditions of their language, and the more time has passed since a word was first borrowed, the harder it is to detect it from its external characteristics alone [15].

Although we expect some limitations of language-internal evidence for lexical borrowing it seems worthwhile to test to what degree it could be employed for automated borrowing detection approaches in computational comparative linguistics. Assuming that the strongest language-internal evidence for lexical borrowings can be found in the phonology and phonotactics of borrowed words, all that we need to do in a computational approach to monolingual borrowing detection is to derive computational models of phonology and phonotactics from annotated wordlists of a given language and then calculate to which degree a word resembles a typically inherited or a typically borrowed word.

To model the phonology and the phonotactics of a language, we make use of different *lexical language models*. Assuming that a *language model* refers to "any system trained only on the task of string prediction, whether it operates over characters, words or sentences, and sequentially or not" [16], our *lexical language models* are specific cases of language models derived from lexical data typically provided in the form of a wordlist, with words being represented by phonetic transcriptions. Having trained lexical language models for inherited and borrowed words with the help of a given annotated wordlist representing a given language variety in a supervised learning setting, we can then try to measure to which degree words that were not used to train a given model can be classified as either being inherited or borrowed.

In this study, we test how well three different lexical language models—one non-sequential model based on a support vector machine, and two sequential models, one based on Markov chains and one based on recurrent neural networks—perform in detecting borrowed words. We apply our models to the World Loanword Database [17], a large, cross-linguistic sample of wordlists in which borrowed words are annotated, which we considerably improved by adding harmonized phonetic transcriptions instead of the original orthographic representations of word forms.

We perform a series of experiments with real and some with augmented artificial data. While we find borrowing detection results are unsatisfying for many language varieties, they become more promising (a) with increased amounts of borrowings, and (b) when a higher rate of borrowings goes back to a single donor language. Overall, the recurrent neural network method performs better, although the differences in comparison with the other methods are small.

## State of the art

Although the detection of borrowed words is one of the major tasks in historical language comparison, the classical, non-computational techniques which linguists use to identify borrowings have never been properly formalized or explicitly described [10]. As mentioned before, classical linguists make extensive use of *proxies* to assess whether or not a given word

has been borrowed. While most of the evidence linguists employ to detect borrowed words is based on the comparison of *several* languages, conflicts in phonology and phonotactics are also routinely used for borrowing detection, specifically when dealing with recent borrowing events.

Similar to the prevalence of multilingual approaches to borrowing detection in classical historical linguistics, most recent attempts to detect borrowings automatically have also been based on comparative rather than monolingual evidence. Various authors have tried to detect borrowings by searching for phylogenetic conflicts [18–24]. Other approaches identify similar words in unrelated languages [25–27]. Occasionally, authors have tried to detect borrowings by relying on the idea that some words can be more easily borrowed, because of the meanings they express [28]. While the detection of words borrowed between unrelated languages seems to work relatively well [27], all other approaches that have been proposed in the past, have never been rigorously tested.

In contrast to multilingual approaches to borrowing detection, monolingual approaches in which borrowings are identified by relying on the (annotated) data of one language alone, have been rarely applied so far, and the rare exceptions we know of, involve very particular settings for individual languages, as opposed to generic approaches that could be generally applied [29, 30].

Although—to our knowledge—language models have not yet been used to identify borrowings in exclusively monolingual wordlists, the idea of using lexical language models for specific tasks in comparative linguistics is not new. Language identification, for example, which seeks to identify the natural language in which a given document is written [31], shows certain similarities with the task of monolingual borrowing detection. Distinguishing foreign words within a paragraph or sentence is similar to the problem of detecting recently borrowed words in a wordlist.

## Materials and methods

### Materials

We use the multilingual wordlist collection provided by the World Loanword Database (WOLD) [17], which we modified by adding harmonized phonetic transcriptions. Each of the 41 wordlists in this collection provides translation equivalents for 1,460 distinct concepts (see the Concepticon resource for details on this concept list [32]). Since translations may lack or one concept may have been represented by more than one word form, the resulting wordlists comprise between 956 and 2,558 word forms. While word forms were provided in orthographic form or phonological transcriptions in the original data, we added phonetic transcriptions which follow the unified *Broad IPA* transcription system proposed by the Cross-Linguistic Transcription Systems reference catalog [33, 34] with the help of *orthography profiles* [35] manually compiled by reading the relevant literature for each language. Orthography profiles can be best thought of as a specific look-up table, which allows to convert transcriptions from one orthography into another one (compare the presentation in Wu et al. [36] for details); while such assisted transcription can introduce noise in the data, no comparable lexical database with transcriptions and loanword annotation was available. Each word form is given a so-called *borrowed score*, indicating the rating of a linguistic expert that the item was borrowed on a five-point scale. To make sure that we only consider clear-cut borrowings in our tests, we treated as borrowed only the words which were labeled as *clearly borrowed*.

The derived database with phonetic transcriptions for all 41 wordlists was curated with the help of the CLDFBench toolkit [37], which allows for a convenient, test-based data curation workflow in which the resulting dataset is offered in the formats recommended by the Cross-

**Table 1. Illustrative subset with the most salient information in the data.**

| ID | Language | Concept | Value | Segments | Borrowed |
|---|---|---|---|---|---|
| 1 | Swahili | WORLD | dunia | ɗ u n i a | True |
| 45 | TarifiytBerber | VALLEY | tizi | θ i z i | False |
| 481 | English | CALM | calm | k ɑː m | True |
| 992 | Mapudungun | FOAM | tronün | tʂ o n ɨ n | False |

Linguistic Data Formats initiative (CLDF, https://cldf.clld.org [38]). These format specifications have proven very useful in the past, as they allow not only for a quick aggregation of data from different sources [39], but also for their convenient integration in computational workflows [36]. An illustrative subset of the data, as stored in memory for training and evaluation, is provided in Table 1.

For testing purposes, we created an additional German wordlist, taken from an etymological dictionary of German [40], with phonetic transcriptions added with modifications from the CELEX database [41]. While the enhanced WOLD database has been curated on GitHub (https://github.com/lexibank/wold) and archived with Zenodo [42], the German wordlist, available as a stand-alone tabular file in the package we wrote for monolingual borrowing detection, represents an older version of a refined wordlist that, combining additional sources [43], has been published separately [44].

## Lexical language models

For the purpose of testing how well borrowed words in a wordlist can be detected through language-internal information alone, we employ three different lexical models which reflect unique characteristics of phonological and phonotactic clues which can be used to identify borrowings. The *Bag of Sounds* method represents words internally as a set of the sounds of which they consist, the *Markov Model* represents words by their sound *n*-grams, and the *Neural Network* represents words in the form of sequences of learned vector representations of sounds.

We perform borrowing detection on each wordlist individually, modeling word expectedness with Bag of Sounds, Markov Model [45], and Neural Network [46] methods. The Bag of Sounds is a baseline method, which uses a support vector machine to directly detect borrowings based only on the set of sounds. The Markov Model and Neural Network produce sequential sound segment probability estimates, which we transform into word entropies and use to predict borrowed words. The Markov Model serves as the standard approach and the Neural Network as an improved alternative to borrowing detection with entropy methods. The Markov Model and Neural Network methods focus on phonotactics, while the Bag of Sounds method focuses on phonology.

**Bag of Sounds.** Since the word forms in our data are available as harmonized phonetic transcriptions, it is straightforward to represent each word form in a given language as a vector indicating the presence and absence of distinct sound segments. Since the order of these sound segments is not important, and neither is their frequency considered, this vector can be thought of as a simple bag of sounds, in which the sounds making up a given word form are represented as a set. The task of distinguishing borrowed from inherited words can then be pursued with the help of a support vector machine with a linear kernel [47, 48]. The support vector machine identifies the plane which optimally separates inherited from borrowed words based on the set of sound segments. The Bag of Sounds method does not consider the order or the frequency of elements in a given sound sequence, and we did not expect it to perform

extraordinarily well in all languages in our sample. The advantage of the model is that it is simple and fast in application. It also provides a baseline for those cases where peculiar sounds provide enough information to identify a given borrowed word.

**Markov Model.** An $n - 1$ order Markov Model, emits a sound segment with probability dependent on the $n - 1$ previous sound segments (an n-gram model). The product of sound segment probabilities estimated by the Markov Model are transformed into per sound segment word entropies which are then used in borrowing detection.

We use a second order Markov Model, a 3-gram model, from the Natural Language Toolkit (NLTK) [49]. In the second order model, the emission probability, $P(c_k|c_{k-2}^{k-1})$, is conditioned on the previous 2 sound segments. The second order Markov Model is local with longer range effects resulting from the second order probabilistic process.

We can approximate the probability of a sequence of $n$ sound segments that make up a word, $P(c_1^n)$, by the product of the $n$ second order conditional probabilities:

$$P(c_1^n) \approx \prod_{k=1}^{n} P(c_k|c_{k-2}^{k-1}).$$

We transform word probabilities to a per sound segment word entropy,

$$H(w) = -(1/n)\log P(c_1^n),$$

which typically exhibits a smooth distribution with moderate right skew for wordlists. The second order model with a sound segment vocabulary size $V$ requires $V^3$ probability parameters for sound segment emission probabilities conditioned on the previous two sound segments.

With wordlists of just 1,000 to 2,500 word forms and a typical sound segment vocabulary size of $V \approx 50$, estimating $50^3 = 125,000$ parameters by maximum likelihood would cause sparse parameter estimation with problems of both undefined conditional probabilities and overfitting. We use interpolated Kneser-Ney smoothing to accommodate unseen tri-grams, reduce overfitting, and reduce the number of estimated parameters to less than the $V^3$ required under maximum-likelihood.

**Recurrent Neural Network.** Recurrent Neural Networks provide word length order conditioning via the recurrent layer with memory. Word probabilities are expected to be better estimated, i.e., better approximating human performance, than for the Markov Model, as we can infer from early work of language modeling by [46] and more recent work with transformer language models [50].

Conditional sound segment emission probabilities are dependent on and estimated from all earlier sound segments of the current word:

$$P(c_k|c_1^{k-1}) = f(c_{k-1}, \dots, c_1).$$

We can approximate the probability of a sequence of $n$ sound segments that make up a word, $P(c_1^n)$, by the product of the $n$ corresponding conditional probabilities:

$$P(c_1^n) \approx \prod_{k=1}^{n} P(c_k|c_1^{k-1}).$$

Word probabilities are again transformed to a per sound segment word entropy.

$$H(w) = -(1/n)\log P(c_1^n).$$

The challenge and advantage of the recurrent Neural Network method is in the estimation of the conditional sound segment probabilities, with the function $f(c_{k-1}, \dots, c_1)$, using a more

complex architecture but with fewer parameters (Fig 1) than the second order Markov model. Sparse indicator vectors, $c_k$, representing sound segments are transformed into dense real input vectors, $x_k$. In the recurrent layer, input vectors, $x_k$, and prior hidden state vectors, $h_{k-1}$, are linearly transformed and passed through a *tanh* activation function to produce current hidden state, $h_k$, and output, $o_k$, vectors. Resulting output vectors are linearly transformed in a dense output layer of logits, $y$, representing possible output segments. The *softmax* activation function transforms logit values $y_k$ into sound segment probability estimates,

$$\hat{P}(c_n | c_{n-1}, \ldots, c_1) = e^{y_{c_n}} / \sum_k e^{y_k}.$$

While the recurrent Neural Network model requires a high baseline number of parameters given its embedding length and recurrent layer length, the growth in number of parameters is just linear with the vocabulary size. As a result, the number of parameters in the Neural Network is on the order of 10,000, and this does not change much with the vocabulary size. Furthermore, the number of parameters does not increase with word length in sound segments even though the conditioning is on all previous sound segments.

We implement our recurrent Neural Network in Tensor-Flow 2.2 [51] and parameterize the model to permit ready changes in architecture, regulation, and fitting parameters during experimentation. The configuration used in this study is shown in Fig 1. Neural network models, even with just thousands of parameters, may suffer from substantial variance between training and test due to overfitting, especially when the amount of training data is comparatively small as in this case. We apply methods of dropout and l2 regulation to reduce overfitting.

## Decision procedures

Models are trained on labeled data and then used to predict whether unlabeled test words are inherited or borrowed. The Bag of Sounds method directly decides which test words are

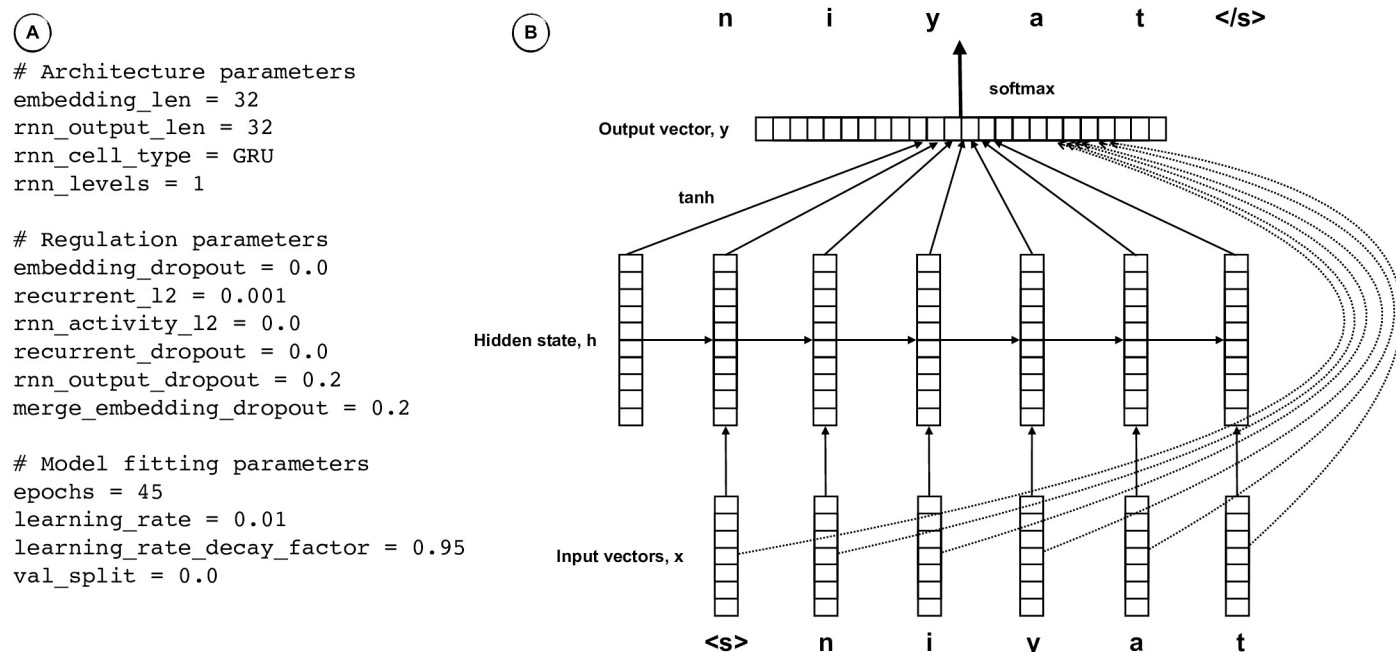

```
A
# Architecture parameters
embedding_len = 32
rnn_output_len = 32
rnn_cell_type = GRU
rnn_levels = 1

# Regulation parameters
embedding_dropout = 0.0
recurrent_l2 = 0.001
rnn_activity_l2 = 0.0
recurrent_dropout = 0.0
rnn_output_dropout = 0.2
merge_embedding_dropout = 0.2

# Model fitting parameters
epochs = 45
learning_rate = 0.01
learning_rate_decay_factor = 0.95
val_split = 0.0
```

**Fig 1. Recurrent Neural Network—Lexical model.** (A) Configuration parameters. (B) Model architecture.

borrowed. Both Markov Model and Neural Network methods estimate test word entropies from dual models trained on inherited and borrowed words separately.

We assume that for a model trained on inherited words, the entropy estimates for unobserved inherited words will be less than for borrowed words. Similarly, for a model trained on borrowed words, entropy estimates for unobserved borrowed words will be less than for inherited words. Words are designated as inherited or borrowed depending on which of the models has the lesser entropy. The choice of the model with the lesser entropy can be expressed as the difference of entropies compared to a critical value, in this case zero:

$$borrowed = (entropy(w)_{inherited} - entropy(w)_{borrowed}) > 0.$$

## Assessing detection performance

We assess detection performance using *precision*, *recall*, and *harmonic mean* (*F1 score*), as well as *accuracy* measures based on frequency counts of borrowing detection by true borrowing status as defined in Table 2. Following [52], *precision* is the proportion of true positive borrowings out of all detected positives,

$$precision = tp/(tp + fp),$$

*recall* is the proportion of true positive borrowings out of all borrowings,

$$recall = tp/(tp + fn),$$

*F1 score* is the harmonic mean of precision and recall, and

$$F1 = (2 * precision * recall)/(precision + recall),$$

*accuracy* is the proportion of all detections that are correct,

$$accuracy = (tp + tn)/(tp + fp + fn + tn).$$

We consider F1, since it combines both precision and recall, as the primary measure. Accuracy does not specifically focus on borrowing detection and is of secondary importance.

## Implementation

Methods for borrowing detection and evaluation have been implemented in the form of the Python PyBor package and have been published along with supplemental information accompanying this study [53]. The Python package contains the code, access to data, and examples that replicate all studies here presented and illustrate how to perform new analyses.

## Experiments with results

We run several experiments as follows. First, we simulate detection of recent borrowings from artificially seeding wordlists with various proportions of words from a foreign language.

**Table 2. Frequency counts of borrowing detection by true borrowing status.**

| Borrowing Detection | True borrowing status | |
|---|---|---|
| | **Borrowed** | **Inherited** |
| Positive | tp = true positive | fp = false positive |
| Negative | fn = false negative | tn = true negative |

Second, we test borrowed word detection more realistically by using wordlists without alteration. Third, we perform correlational and regression analyses to diagnose performance as a function of proportion of borrowed words and phonological variables. Fourth, we stratify wordlists by number of borrowed words and presence of a dominant donor language and analyze borrowed word detection by strata. Last, we examine entropy distributions for a few exemplary wordlists to see how the entropy method works.

## Detection of artificially seeded borrowings

To simulate a situation in which foreign words have recently entered a language without being modified by borrowed word nativization processes, we designed an experiment in which the wordlists in our base datasets were artificially mixed with words from another wordlist which was not part of the original WOLD collection. The idea to use "artificially seeded" borrowings instead of borrowings attested in actual language was originally proposed for evaluating methods for lateral gene transfer detection in biology [54], and later tested on linguistic data in order to assess the power of phylogenetic methods for borrowing detection across multiple languages [23]. The advantage of this procedure is that it creates simulated data without requiring the efforts of detailed simulation experiments.

Artificial borrowings were seeded into a wordlist in three steps. We first removed all borrowed words from the wordlist to guarantee that no recent borrowings from other languages could influence the results. We then added inherited words from the additional German list, which we created for testing purposes. Here, we tested three different proportions of borrowed words, 5%, 10%, and 20%, in order to allow to compare different degrees of contact. In a final step, we then split the resulting wordlist into a training and a test set (reserving 80% of the data for training and 20% for testing) and ran the three methods for monolingual borrowing detection, Bag of Sounds, Markov Model, and Neural Network.

The results of this experiment are given in Table 3, where the borrowing detection results are provided in form of *precision*, *recall*, and *F1 scores* for the three different borrowing rates. Fig 2 presents plots for 5% and 10% borrowing rates. Accuracy results, not shown, were all above 0.95 and varied little over methods and rates. Individual results indicating the scores achieved by method and borrowing rate for each language are provided as in S1 Table.

As can be seen from the results, all methods perform well when artificially seeded borrowings amount to 20%. With a borrowing rate of 10%, all methods still achieve F1 scores of more than 0.90, with the Bag of Sounds showing the lowest precision and the Markov Model

**Table 3. Borrowing detection results for artificially seeded borrowings.**

| Method | Rate% | Prec. | Recall | F1 |
|---|---|---|---|---|
| Bag of Sounds | 5 | 0.80 | 1.00 | 0.88 |
| Markov Model | 5 | 0.96 | 0.67 | 0.76 |
| Neural Network | 5 | 0.97 | 0.84 | 0.90 |
| Bag of Sounds | 10 | 0.87 | 0.99 | 0.92 |
| Markov Model | 10 | 0.96 | 0.87 | 0.91 |
| Neural Network | 10 | 0.97 | 0.93 | 0.95 |
| Bag of Sounds | 20 | 0.91 | 0.99 | 0.94 |
| Markov Model | 20 | 0.97 | 0.94 | 0.95 |
| Neural Network | 20 | 0.99 | 0.97 | 0.98 |

Results averaged over all languages for each method and borrowing rate.

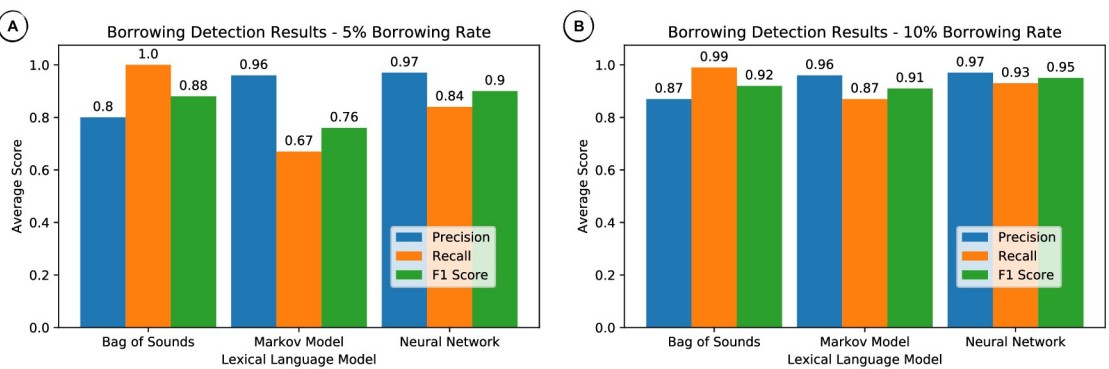

**Fig 2. Borrowing detection results for artificially seeded borrowings.** (A) 5% borrowing rate. (B) 10% borrowing rate.

showing the lowest recall. When borrowings only amount to 5%, we can observe the same trend of low precision for the Bag of Sounds and low recall for the Markov Model. However, while the Bag of Sounds still comes close to the performance of the Neural Network with respect to the F1 score (0.88 vs. 0.90), the Markov Model shows a drastic drop here, resulting from the dramatic loss in recall (0.67).

## Borrowing detection on real language data

Our experiment on artificially seeded borrowings simulated an ideal situation of language contact in which new words were recently introduced into a given language without being adjusted to the recipient language's target phonology. While this experiment provided high scores in our evaluation experiment, the experiment does not allow us to estimate how well the three borrowing detection methods will perform when being exposed to "real" data. For this reason, we designed a second experiment on the WOLD data in their original form. Given that the wordlists are quite small, while specifically Markov Model and Neural Network language models tend to require larger amounts of data, we used cross validation techniques, in which the data are repeatedly partitioned into training and test data and evaluation results are measured for each trial and later summarized. We employed *ten-fold cross validation* for this experiment, where each word list was partitioned into 10 parts, and over 10 successive trials, one part was successively designated the test set while the remaining nine parts were designated the training set. This resulted in 10 separate estimates of borrowing detection performance, with each word appearing once in test sets and nine times in training sets.

Table 4 shows the averages and standard deviations of results (*precision*, *recall*, *F1 score*, *accuracy*) of this experiment for each of our three methods. Fig 3 summarizes the averaged results. Individual results indicating the scores achieved by method for each language are provided as in S2 Table.

As can be seen from the table and the figure, the Neural Network marginally outperforms the Markov Model, while both the Neural Network and the Markov Model clearly outperform the Bag of Sounds. The strength of the entropy-based methods lies in their high precision, while the Bag of Sounds shows the highest recall, but an extremely low precision.

When examining the individual results achieved by each method for each individual language in our sample, one can find a rather huge variation in the results, ranging from results which one may consider as satisfying (such as the performance of the Neural Network on Zinacantán Tzotzil with an F1 score of 0.81) up to extremely bad results (such as the

**Table 4. Borrowing detection results of the cross validation experiment.**

| Method | Statistic | Prec. | Recall | F1 | Acc. |
|---|---|---|---|---|---|
| Bag of Sounds | Mean | 0.286 | 0.578 | 0.349 | 0.843 |
| | Language SD | 0.250 | 0.287 | 0.268 | 0.081 |
| | Pooled SD | 0.078 | 0.226 | 0.088 | 0.030 |
| Markov Model | Mean | 0.678 | 0.521 | 0.578 | 0.828 |
| | Language SD | 0.136 | 0.181 | 0.170 | 0.060 |
| | Pooled SD | 0.114 | 0.088 | 0.082 | 0.034 |
| Neural Network | Mean | 0.697 | 0.546 | 0.603 | 0.844 |
| | Language SD | 0.164 | 0.191 | 0.181 | 0.062 |
| | Pooled SD | 0.100 | 0.082 | 0.072 | 0.030 |

Mean and standard deviation over languages, and pooled standard deviation within languages for each method over all languages.

performance of all methods on Mandarin Chinese, with F1 scores below 0.02). The reasons for the underwhelming results on Mandarin Chinese are twofold. On the one hand, the language barely borrows words directly, but rather resorts to *loan translation*, by which new concepts are rendered with the help of the lexical material in the target language. As a result, Mandarin has the lowest amount of direct borrowings in our sample. On the other hand, Mandarin Chinese (as well as all Chinese dialects and many languages from Southeast Asia) has an extremely restricted syllable structure that makes it impossible to render most foreign words truthfully [55]. As a result, words are usually directly adjusted to Chinese phonotactics when being borrowed and also written with existing Chinese characters, which again further masks their foreign origin [56]. However, this very specific situation also makes it also difficult if not

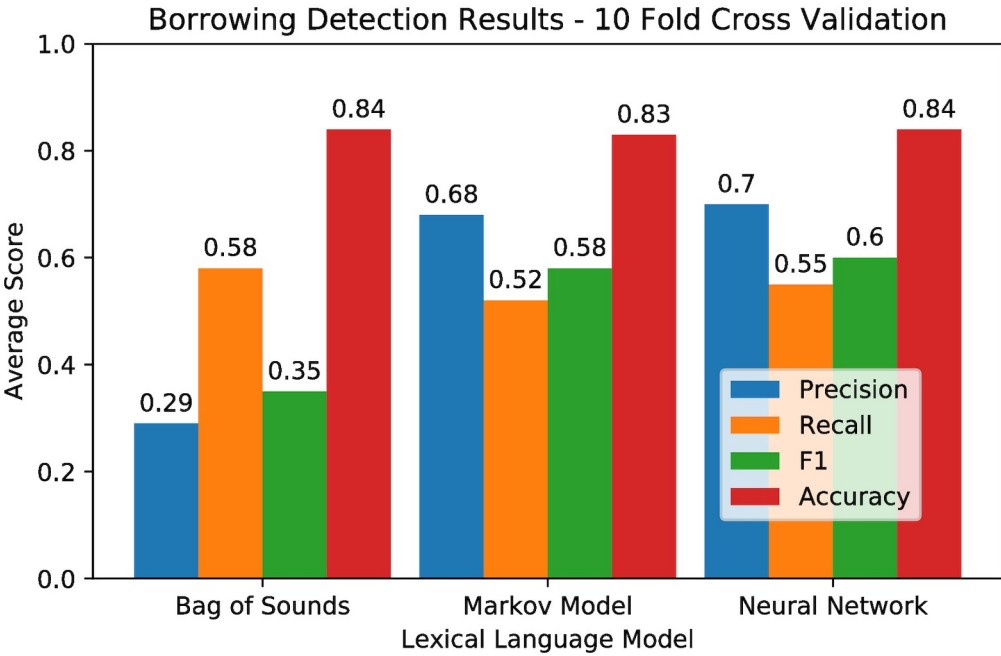

**Fig 3. Results of the cross validation experiment.** Averaged for each model over all languages in our sample.

impossible for most Mandarin Chinese speakers to identify borrowings when considering phonotactic criteria alone.

## Factors that influence borrowing detection

Given that the performance of our supervised borrowing detection methods varied substantially, ranging from poor performance with F1 scores below 0.5, average performance with F1 scores between 0.5 and 0.8, and acceptable performance with F1 scores above 0.8, we performed analyses to assess to which degree certain factors might influence the borrowing detection methods.

In concrete, we computed specific characteristics of each language variety in our sample and then checked to which degree these characteristics correlated with the test performance. As characteristics, we chose the proportion of borrowed words in a given language wordlist, since statistical and machine learning methods perform better with sufficient representation, and the proportions of unique sounds in borrowed words and in inherited words, as potential contributors to prediction performance. A higher proportion of borrowed words corresponded moderately to a lower proportion of unique sounds in inherited words, otherwise characteristics were independent. Statistical analysis, correlational study, matrix plots, and regression, were performed with Minitab® Statistical Software [57]. The correlation results, based on all wordlists in our sample taken from the WOLD database, are reported in Table 5, and accompanied by detailed plots in Figs 4, 5 and 6. We focus on the strength of the relationship between characteristics and borrowing detection rather than statistical significance.

As can be seen from the correlations and the plots, there is a moderately strong to strong positive correlation between the proportion of borrowed words and the evaluation scores for all tests. The effect of proportion of borrowed words appears non-linear for the entropy methods, where less than 5% borrowings has much worse borrowing detection than expected in the linear correlation plot from Figs 5 and 6. For the other factors, the proportion of sounds occurring exclusively in borrowed words, and the proportion of sounds occurring exclusively in inherited words, the results are less clear. While we observe a moderate correlation between the proportion of exclusively borrowed sounds with the recall for the Bag of Sounds, there is an equal or higher correlation with the precision for the other two methods.

In order to further investigate the influence of the three factors on the borrowing detection performance, we further analyzed them by fitting a multiple regression model to them. Our major goal was to check whether exclusively borrowed and exclusively inherited sound proportions can help us explain the methods' performance beyond the overall proportion of borrowed words in each wordlist. We fit a second order regression model to predict F1 scores from our three characteristics using Minitab's forward information criteria for model selection. Regression results are reported in Table 6. Almost 50% of variability in performance is explained for the Markov Model and Neural Network. In both cases proportion of borrowed words and proportion of exclusively borrowed sounds strongly contribute to F1 performance.

**Table 5. Correlations between phonological characteristics and performance of borrowing detection methods.**

| Proportion of | Bag of Sounds | | | Markov Model | | | Neural Network | | |
|---|---|---|---|---|---|---|---|---|---|
| | Prec. | Recall | F1 | Prec. | Recall | F1 | Prec. | Recall | F1 |
| Borrowed words | 0.584 | 0.337 | 0.539 | 0.387 | 0.736 | 0.654 | 0.399 | 0.690 | 0.600 |
| Borrowed sounds | 0.185 | 0.345 | 0.199 | 0.345 | 0.274 | 0.297 | 0.377 | 0.268 | 0.301 |
| Inherited sounds | -0.006 | -0.010 | -0.004 | 0.035 | -0.330 | -0.263 | -0.075 | -0.178 | -0.148 |

All correlations with $|r| \geq 0.33$ are significant at $p < 0.05$.

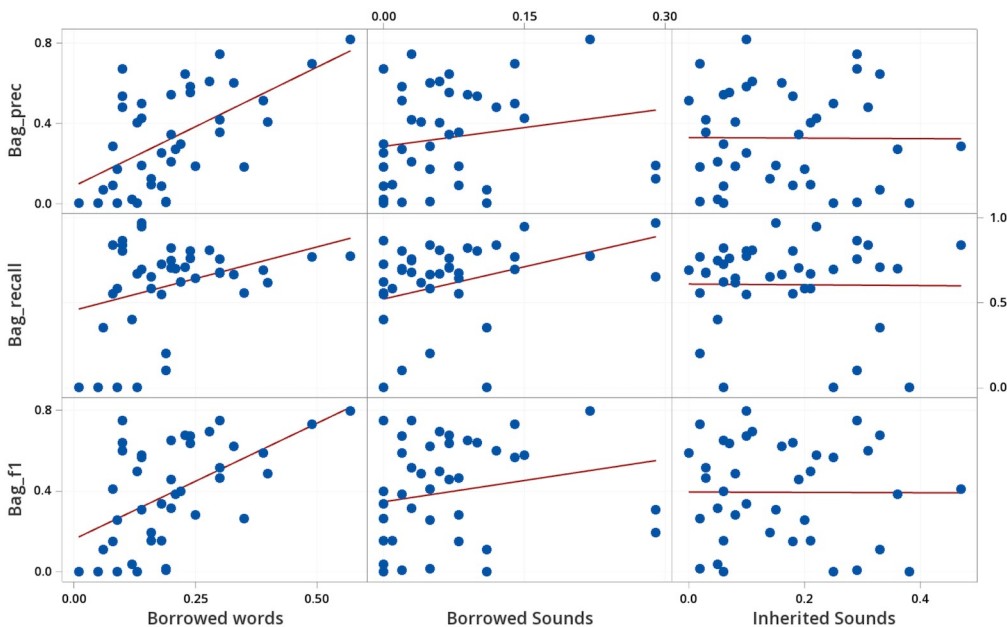

**Fig 4. Determining characteristics that influence the performance of the Bag of Sounds.**

For the Neural Network, exclusively inherited sounds also has a minor impact on F1 performance. The negative coefficients for the squared proportion of borrowed words and exclusively borrowed sounds terms, serve to flatten out F1 performance at 0.8 and higher. Almost 30% of variability in performance is explained for the Bag of Sounds, most of this due to the proportion of borrowed words with a minor impact due to exclusively inherited sounds.

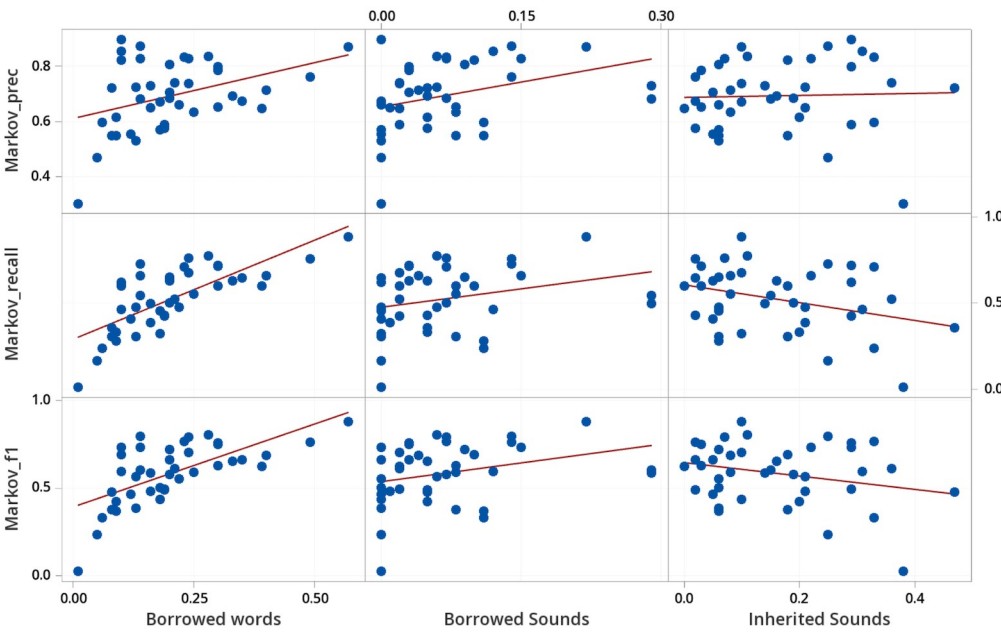

**Fig 5. Determining characteristics that influence the performance of the Markov Model.**

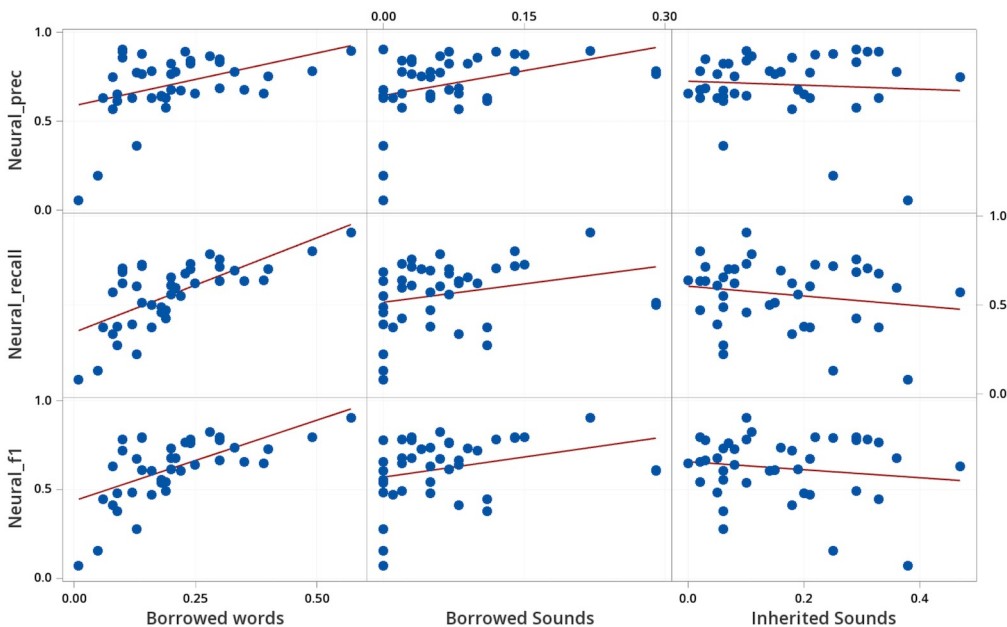

**Fig 6. Determining characteristics that influence the performance of the neural network.**

## Detecting borrowings from a single donor language

Testing our lexical language models on the WOLD data in their entirety could be considered as unfair to the methods, given that we know well that monolingual evidence for borrowing in phonotactics may get lost easily and that the WOLD database was never restricted to recent borrowings alone. Another problem of the data is that the distinction between inherited words on the one hand and borrowings on the other hand is as well a simplifying assumption, since we know that in intensive contact situations borrowings come from a specific donor language. As a result, it seems to be justified to test the three methods for monolingual borrowing detection with the help of more specific experiments in which the task consists in the detection of borrowings when there is a single or dominant language donor, as in intensive contact situations, versus the case when no language donor dominates.

To test whether our methods show an improved performance when there is a dominant language donor as opposed to detecting borrowed words *per se*, we first created two subsets of the WOLD database, one containing languages with 300 and more borrowed words (17 language varieties), and one containing languages with 100 and more borrowed words (37 language varieties). We then searched for "dominant donor languages" in all wordlists in each sample, with dominant donor languages being defined as those donor languages (as identified in the WOLD database) that would account for two-thirds of all borrowings identified for a given language variety. For our sample of language varieties with 300 and more borrowings,

**Table 6. Regression analysis on phonological characteristics that influence borrowing detection F1 scores.**

| Method | Regression model | pred-$R^2$ |
|---|---|---|
| Bag of Sounds | $F1 = -0.040 + 1.53bw + 0.76is$ | 29.9% |
| Markov Model | $F1 = 0.141 + 2.66bw + 2.05bs - 3.38bw^2 - 5.05bs^2$ | 48.8% |
| Neural Network | $F1 = 0.032 + 3.12bw + 2.43bs + 0.43is - 3.93bw^2 - 6.35bs^2$ | 49.9% |

**Table 7. 10-fold cross validation—Dominant versus no dominant donor.**

| Borrowed | Method | Donor | Precision | p< | Recall | p< | F1 | p< |
|---|---|---|---|---|---|---|---|---|
| ≧ 300 | Bag of Sounds | Dominant (8) | 0.536 | .0300 | 0.739 | .0200 | 0.588 | .0400 |
| | | No dominant (9) | 0.308 | | 0.672 | | 0.390 | |
| | Markov Model | Dominant | 0.785 | .0030 | 0.722 | .0020 | 0.749 | .0030 |
| | | No dominant | 0.672 | | 0.585 | | 0.622 | |
| | Neural Network | Dominant | 0.810 | .0002 | 0.722 | .0070 | 0.760 | .0030 |
| | | No dominant | 0.690 | | 0.606 | | 0.642 | |
| ≧ 100 | Bag of Sounds | Dominant (20) | 0.418 | .0030 | 0.737 | .0020 | 0.490 | .0010 |
| | | No dominant (17) | 0.192 | | 0.498 | | 0.252 | |
| | Markov Model | Dominant | 0.762 | .0002 | 0.600 | .0300 | 0.661 | .0060 |
| | | No dominant | 0.639 | | 0.505 | | 0.558 | |
| | Neural Network | Dominant | 0.787 | .0002 | 0.619 | .0200 | 0.685 | .0060 |
| | | No dominant | 0.655 | | 0.523 | | 0.567 | |

this yielded a partition of the data into 8 language varieties for which a dominant donor could be identified and 9 for which none could be found. For the sample of language varieties with 100 and more borrowings, the partition yielded 20 language varieties with a dominant donor and 17 without. We were able to apply results of the 10-fold cross validation study for these two subsets of the data, which we had previously applied to all language varieties in the WOLD database. In order to test whether the observed differences between dominant donor and no dominant donor categories were significantly different, we also performed randomization resampling tests of 5,000 iterations each, using Student's independent $t$ statistic with unequal variances as our test statistic. We report $p$-values from the empirical distribution of $t$ statistics calculated under the hypothesis of no difference due to dominant donor, i.e., dominant and no dominant categories are exchangeable.

As can be seen from the results in Table 7, the performance of all borrowing detection methods improves when the vast majority of the borrowings come from a single donor language. The performance also improves, as we saw previously, with more borrowed words. While performing worse than the other two methods, the Bag of Sounds method shows a strong increase in performance, which is mostly owed to a strong increase in precision, when most borrowings come from a single donor language.

## Comparing entropy distributions

The Markov Model and the Neural Network methods estimate word entropy on a per sound basis given the inherited or borrowed words on which they are trained. Models trained on inherited words should estimate lower entropies for inherited words, and models trained on borrowed words should estimate lower entropies for borrowed words. However, since words are borrowed over time and potentially also from various donor languages, using a single language model for borrowed words is not always optimal.

Our decision procedure for the Markov Model and the Neural Network methods requires the comparison of competing entropies for a given word, the entropy of the lexical language model derived from inherited words and the entropy of the lexical language model derived from borrowed words. If the difference between the entropies is greater than zero, we designate the word as borrowed, and if it is smaller than or equal to zero, we designate the word as inherited.

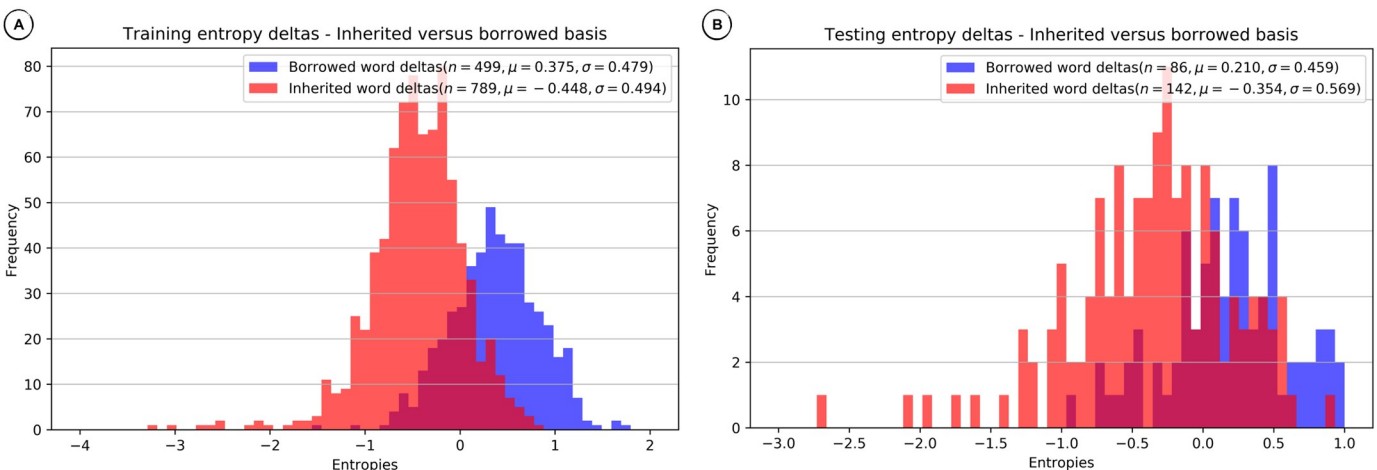

**Fig 7. Distribution of entropy differences for English—Neural Network method.** (A) Training (85%) entropy deltas. B) Testing (15%) entropy deltas.

In order to investigate the *discriminative force* of this procedure, it is useful to compare entropy difference distributions of inherited and borrowed words for a given language variety. The distributions for training and test data from the English wordlist in the WOLD database are shown in Fig 7. While there is a certain overlap between entropy difference distributions for inherited and borrowed words, the problem of discriminating between them based on entropy differences seems tractable, and we can assume that improvements in entropy estimation would have an immediate benefit on prediction.

Since both the Markov Model and the Neural Network performed considerably well on Imbabura Quechua, a Quechua language spoken in Ecuador, with an F1 score above 0.8, it is not surprising that we find a good separation between the entropy difference distributions for inherited and borrowed words, as shown in Fig 8.

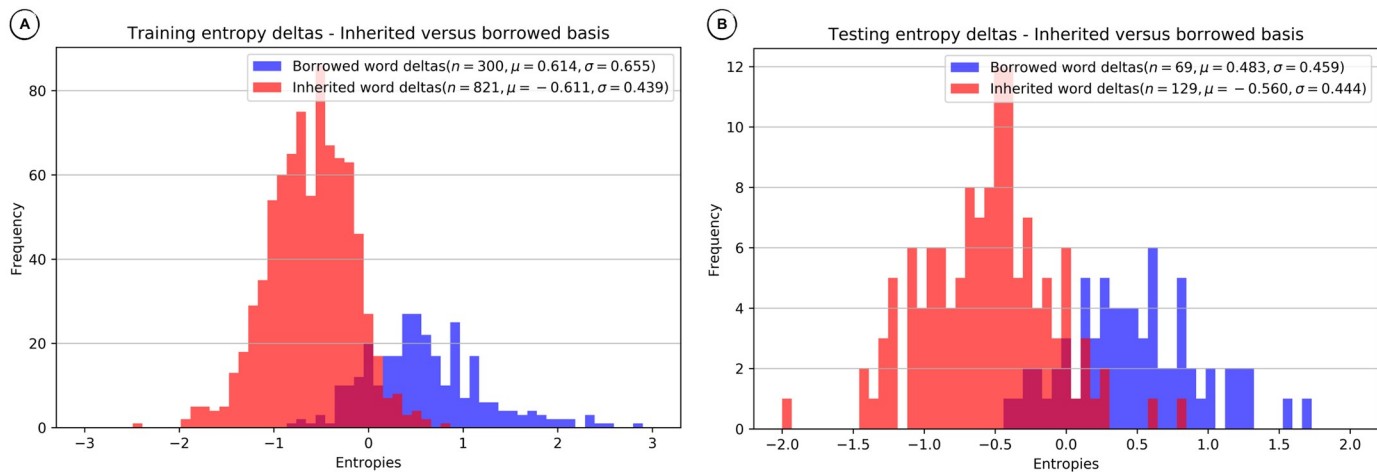

**Fig 8. Distribution of entropy differences for Imbabura Quechua—Neural Network method.** (A) Training (85%) entropy deltas. (B) Testing (15%) entropy deltas.

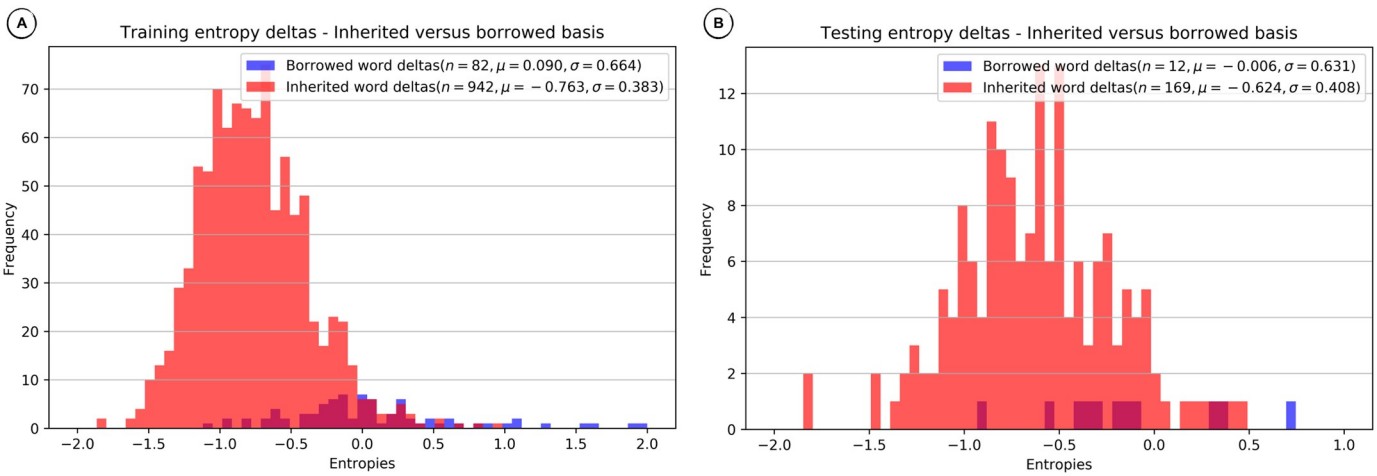

**Fig 9. Distribution of entropy differences for Oroqen—Neural Network method.** (A) Training (85%) entropy deltas. (B) Testing (15%) entropy deltas.

Neither method performed very well on Oroqen, a Northern Tungusic language spoken in the Mongolian region of the People's Republic of China, with F1 scores below 0.36. Consequently, as can be seen in Fig 9 the entropy difference distributions for inherited and borrowed words are not well separated.

This strong relationship between the distribution of entropy differences and borrowing detection, indicates a tactic for improving monolingual lexical borrowing detection—increase the separation of difference distributions for inherited versus borrowed words. An examination of our sample cases reveals: 1. English and Imbabura Quechua, even though there were substantial borrowings, have reduced separation between inherited and borrowed word difference distributions for testing, resulting in reduced discriminative power, and 2. Oroqen, with few borrowings, has almost no separation between inherited and borrowed word distributions for testing, resulting in little discriminative power. Identification of problems permits trying to solve them, such as through improved controls for training of Neural Networks, and by obtaining more borrowings, real or simulated, for training.

## Discussion

### Artificially seeded borrowings

In our artificially seeded borrowings experiment, we simulated very close, intensive, and recent language contact, where borrowed words were transferred without alteration. All methods performed well when the proportion of artificially borrowed words was high, and degraded differently when borrowings decreased.

While the Bag of Sounds outperformed the other two methods regarding recall, the Markov Model and, especially, the Neural Network outperformed the Bag of Sounds method in precision. Since the core strategy of the Bag of Sounds lexical language model is to identify borrowed words by their specific sounds, while the order of sounds itself is ignored, it is not surprising that the method performs better in identifying artificially seeded borrowings, i.e., better recall, since the direct transfer of words from one wordlist to another wordlist, as it was done in our experiment, will always introduce a larger number of sounds which were not present in the recipient wordlist prior to the transfer. In contrast, taking into account the order of

sounds, gives the Markov Model and Neural Network a tremendous advantage at ruling out unseen borrowed word sound sequences resulting in uniformly high precision.

## Borrowing detection on real language data

In our real language borrowing detection experiment, we performed a 10-fold cross validation of borrowing detection across unaltered WOLD wordlists. Here, the Neural Network performed marginally better than the Markov Model, and both of them performed much better than the Bag of Sounds method.

A major factor favoring the Neural Network seems to be that it includes conditional dependencies from all previous sound segments, without having to explicitly estimate numerous extra parameters for this dependency. Similar to the previous experiment, the Bag of Sounds method showed a high recall, but suffered from a low precision as well. So while the Bag of Sounds *suspects* considerably many words of being borrowings, it does not necessarily always pick the right ones and shows a rather high rate of false positives, as can be seen from the low rates of precision. In contrast, the Markov Model and the Neural Network methods show a lower recall, but also a much higher precision. They are therefore much more *conservative* than the Bag of Sounds method. When the overall proportion of borrowed words in wordlists is small, all models perform poorly. This is not surprising, since low borrowing proportions make it difficult to learn the phonotactics or phonology of borrowed words, if these can be identified at all. It is also not clear to which degree trained linguists would be able to identify borrowed words in the respective languages and even less so over entire wordlists instead of just recent borrowings, if they were given only monolingual information alone.

## Factors influencing borrowing detection

Given the disappointing results with real language data, we tried to determine the major factors that influence the performance of borrowing detection methods. Besides proportion of borrowings, we thought that proportions of sounds occurring exclusively in borrowed words and exclusively in inherited words might impact borrowing detection, especially for the Bag of Sounds method, which explicitly deals with sounds, while ignoring phonotactic aspects.

While the effect of the proportion of borrowed words was remarkable, showing a strong linear increase in performance for all methods when the proportion of borrowed words was 5% and more, the impact of the proportions of sounds occurring exclusively in borrowed words and sounds occurring exclusively in inherited words surprised us. Based on the regression model, increased frequency of sounds occurring exclusively in borrowed words resulted in improved performance of both Markov Model and Neural Network methods, but not for the Bag of Sounds method, while frequency of exclusively inherited sounds had little impact on any of the methods. It seems that modeling phonotactics with Markov Model and Neural Network methods also takes good advantage of the simple occurrence of borrowed sounds in words too. With respect to the Bag of Sounds method, what we may have overestimated was that—even if a given language has many sounds occurring exclusively in borrowed words— this does not mean that these sounds need to occur in each and every borrowed word. Thus, while the presence of specific sounds may be a powerful indicator of a borrowing or an inherited word, this evidence may be too sparse in comparison with the full lexicon of a given language.

## Detecting borrowings from a single donor language

Since we create lexical language models for borrowed and inherited words, it is straightforward to question why our basic approach would treat all borrowed words as if they come from a

single donor language. While it may hold for specific contact situations that a given language is heavily influenced by one single, dominant donor language, it is also possible that borrowings form distinct layers in the lexicon of a given language, reflecting borrowings from different donor languages and different times. If the majority of the borrowings attested in a given language stem from a single donor, however, we would assume that our lexical language model approaches to monolingual borrowing detection would perform better, since the donor language which we access through the recipient language would provide a much more coherent and consistent picture than would a mix of words from different donor languages.

We therefore systematically tested whether the performance of our methods would increase for those wordlists in our sample for which a dominant donor language could be identified. Our assumption, that the methods should show an increased performance for languages with a dominant donor language were largely confirmed, as reflected in substantially increased F1 scores of ≈0.75 for the Markov Model and the Neural Network methods in cases of high contact with more than 300 borrowings. While we still consider the overall performance of the monolingual borrowing detection disappointing, this experiment reflects the importance of having a consistent sample of the donor language when dealing with monolingual borrowing detection.

### Comparing entropy distributions

Our final evaluation was intended to demonstrate how the Markov Model and Neural Network methods discriminate between inherited and borrowed words. We showed how plots of the distribution of entropy differences between competing inherited and borrowed word models served to explain borrowing detection results. When comparing the distributions of entropy differences, we found that in those cases where the proportion of borrowings was small, the discriminative force of the word entropy differences seemed to drop drastically for testing. Even when the proportion of borrowings seemed adequate for training we saw a reduction in discriminative force for testing due to reduced separation of inherited and borrowed word entropy difference distributions. This provided additional evidence that monolingual borrowing detection heavily depends on the presence of a large enough proportion of borrowed words, and also that modest improvements might be possible with improved training controls.

### Conclusion

We presented three supervised methods, Bag of Sounds, Markov Models, and Neural Networks, for the detection of borrowings in monolingual wordlists. These methods are based on lexical language models and are intended to model specific aspects of phonology and phonotactics in the lexicon of spoken languages. Assuming that phonological and phonotactic properties of words in the lexicon of a spoken language can provide enough clues to identify borrowings by language-internal comparison of words alone, we designed workflows in which the lexical language models could be trained with monolingual wordlists in which borrowings are annotated and then used to detect borrowings when being confronted with so far unobserved words.

While tests on artificially seeded borrowings showed promising results, tests on real wordlists taken from the WOLD database revealed a rather disappointing performance for all three methods. Consecutive attempts to identify the potential reasons for this mediocre performance revealed two main factors that considerably influence how well the methods performed, namely (1) the amount of borrowings in a given language variety, and (2) the uniformity of the borrowings in a given language variety, as reflected in the presence of a dominant donor

language. While the first factor reflects the importance of having enough training data when working in supervised learning frameworks, the second factor reflects more specific linguistic conditions of monolingual borrowing detection. Our methods identify borrowings primarily from phonological and phonotactic clues, and perform better in those cases where the words' properties are coherent and consistent. This is generally the case for inherited words, and also for words that were borrowed uniformly from the same donor language.

While our results do not recommend any of the three methods represented here as a replacement for previously proposed methods for borrowing detection, we believe that the methods we created offer a valuable and promising baseline for the further exploration of monolingual approaches to borrowing detection. We are even convinced that they may be useful in some concrete applications. Given that we know that our methods rely heavily on sufficiently large samples of training data, our methods may be useful especially for those studies in which borrowed words or sentences need to be identified in large amounts of data, preferably in situations where borrowings are considerably young. Here, especially, larger linguistic corpora could be analyzed and tagged for inherited and borrowed words.

But even if future research shows that our attempts to model phonology and phonotactics with the help of language models in a supervised framework for monolingual borrowing detection cannot be improved any further, we consider it worthwhile to share our results along with the software and the data we used to create them, since this may—in the worst case—save those colleagues who might want to test the same idea some precious time.

Additionally, we think that—given that by now no single method for borrowing detection has been proposed that exhibits satisfactory performance—our methods add to the growing pool of automated approaches to borrowing detection which could ideally be later combined into an *integrated workflow* in which evidence from multi-lingual sources can be combined to form a unified picture of language contact.

## Supporting information

**S1 Table. Detection results by language for seeded borrowings.** Borrowing rates of 5%, 10%, and 20%.
(PDF)

**S2 Table. Ten-fold cross validation of detection results by language for WOLD wordlists.** Cross-validation means and standard deviations.
(PDF)

## Acknowledgments

We thank Mei-Shin Wu for work on the White Hmong and Mandarin profiles used to convert WOLD word forms to IPA sound segments. We also thank the Chana team (promoting technologies for indigenous languages of Peru) of the Pontificia Universidad Católica del Perú (PUCP) for their help and encouragement.

## Author Contributions

**Conceptualization:** John E. Miller, Tiago Tresoldi, Roberto Zariquiey, César A. Beltrán Castañón, Johann-Mattis List.

**Data curation:** Tiago Tresoldi, Natalia Morozova, Johann-Mattis List.

**Formal analysis:** John E. Miller, Tiago Tresoldi, Johann-Mattis List.

**Funding acquisition:** César A. Beltrán Castañón, Johann-Mattis List.

**Investigation:** John E. Miller, Tiago Tresoldi, Johann-Mattis List.

**Methodology:** John E. Miller, Tiago Tresoldi, Johann-Mattis List.

**Project administration:** Roberto Zariquiey, César A. Beltrán Castañón, Johann-Mattis List.

**Resources:** Johann-Mattis List.

**Software:** John E. Miller, Tiago Tresoldi, Johann-Mattis List.

**Supervision:** Johann-Mattis List.

**Validation:** John E. Miller, Tiago Tresoldi, Johann-Mattis List.

**Visualization:** John E. Miller.

**Writing – original draft:** John E. Miller, Tiago Tresoldi, Johann-Mattis List.

**Writing – review & editing:** John E. Miller, Tiago Tresoldi, Roberto Zariquiey, César A. Beltrán Castañón, Johann-Mattis List.

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
