## [Decision Letter · Decision Letter 0]

1 Oct 2020

PONE-D-20-27304

Using lexical language models to detect borrowings in monolingual wordlists

PLOS ONE

Dear Dr. Miller,

Thank you for submitting your manuscript to PLOS ONE. After careful consideration, we feel that it has merit but does not fully meet PLOS ONE’s publication criteria as it currently stands. Therefore, we invite you to submit a revised version of the manuscript that addresses the points raised during the review process.

All three reviewers have some good suggestions that you should take into account. For Reviewer 1 it was a stumbling block that the paper seems to try to model a native speaker's ability to identify loanwords. Given that it doesn't do a good job at that the verdict was a rejection. But it seems that the paper is really about the extent to which it is possible for a computer to identify loanwords given information about the target language only. If you clearly spell out that focus and downplay the importance of discussion about what native speakers can and cannot then you might avoid some confusion.

We look forward to receiving your revised manuscript.

Kind regards,

Søren Wichmann, PhD

Academic Editor

PLOS ONE

Journal Requirements:

2. Please note that in order to use the direct billing option the corresponding author must be affiliated with the chosen institute. Please either amend your manuscript or remove this option (via Edit Submission).

3. We note you have included a table to which you do not refer in the text of your manuscript. Please ensure that you refer to Table 5, 7, 8, 9 and 10 in your text; if accepted, production will need this reference to link the reader to the Table.

Reviewers' comments:

Reviewer's Responses to Questions

**Comments to the Author**

1. Is the manuscript technically sound, and do the data support the conclusions?

Reviewer #1: Partly

Reviewer #2: Yes

Reviewer #3: Yes

2. Has the statistical analysis been performed appropriately and rigorously? 

Reviewer #1: No

Reviewer #2: Yes

Reviewer #3: Yes

3. Have the authors made all data underlying the findings in their manuscript fully available?

Reviewer #1: Yes

Reviewer #2: Yes

Reviewer #3: Yes

4. Is the manuscript presented in an intelligible fashion and written in standard English?

Reviewer #1: Yes

Reviewer #2: Yes

Reviewer #3: Yes

5. Review Comments to the Author

Reviewer #1: The article is a rather mechanical application of several machine-learning algorithms, two of them severely outdated and none of them state-of-the-art, to what is essentially a non-task. The motivation for the experiments---that speakers of different languages (cited publications refer to Russian and Korean in particular) are good at identifying borrowings---is way too slim.

Firstly, this only applies to very recent borrowings (Russian, just like English, is saturated with older borrowings, which are undetectable by native speakers).

Secondly, this does not provide a cross-lingual baseline against which to compare the performance of ML algorithms.

Thirdly, as the authors point out themselves, this is not how borrowings are detected in the historical-comparative literature (where the principle of irregular sound correspondences is the only one of real standing; of course, this principle is rather hard to automate because one has to establish the correspondences manually to begin with).

It is hard to grasp what exactly the study is trying to show. It could have been construed as an attempt to model discriminative abilities of native speakers, in which case the focus should have been on how the neural net discriminates between native and borrowed words (cf. the abundant recent literature on what BERT might know about syntax, etc.). Instead all the models are treated as black boxes, and the analysis boils down to identifying situations in which they perform better or worse. This may have been of interest had the proposed method been of practical use. Some possible applications are listed in the conclusion ("studies in which borrowed words or sentences need to be identified in large amounts of data"; "[work] on code switching, where multilingual language users switch between different varieties based on sociolinguistic contexts"); however, the fact that the proposed methodology can help there itself needs to be tested against appropriate baselines and competing approaches.

Reviewer #2: The authors introduced a new approach to automatic loanword detection in the field of computational historical linguistics (CHL). While most of the existing methods aim at identifying loanwords in multilingual wordlists, the attempt of this paper is to identify borrowings using a monolingual approach. The authors use the WOLD database, which is the only database containing loanword information along with information about the donor language and loaned status. What I really liked about the work is that the three methods build on another: the Bag of Sound model using an SVM is the simplest model, integrating only the phonology of the words without considering the order and frequency of the sounds; the Markov Model is a tri-gram model relying on the two previous sounds in the word; the recurrent neural network also relies on the phonotactics of the word, taking all previous sounds into account. The authors perform two experiments, one on artificially seeded borrowings and one on the “real” WOLD data. Since the promising results on the simulated data could not be obtained by the experiments using the WOLD data, the authors made additional experiments in order to explain the performance of the methods, which was achieved. Although, the results and the performance are not satisfying, the three introduced methods along with the lexical language models open a new perspective for further research, especially the recurrent neural network, serves as basis for improvements and further explorations in the field of automatic loanword detection.

Two things I really liked about the manuscript are the detailed explanation of the methods and the representation of the data, which help the reader to get clear insights in the evaluation of the different methods. The statistical evaluation are carried out and explained in detail. The authors made the effort to perform additional analyses to explain the performance of the methods in the two experiments. The statistical evaluation are carried out in a rigorous way, giving some explanations according to the performance and the results.

The process of nativization plays a crucial role in the motivation of the approach introduced in the first chapters, however it was not revive in more detail in the conclusion. The detection of loanwords using the proposed methods depends highly on the data and the adjustment status of the word in the recipient language. The methods might not identify older loanwords or loaned words from related languages, which show no clear differences in the phonology. This issue was not discussed in detail in the conclusion, but could be one of the reasons of the poor performance of the methods.

In addition, the automatically derived IPA transcriptions of the words from the WOLD database could lead to noise in the analyses, depending on correctness of the transcription. However, since no other database is available containing loanword annotations along with the donor language and loanword status, compromises need to be made.

The data is completely available online. Within the data, the additional German wordlist used for the artificially seeded borrowings is not identifiable at first sight. I would encourage the authors to provide the list in a format like csv and allocate it at first sight in the python package. Additionally, as a small notice, on p.6 the authors wrote that the German word list and the software package are available on GitHub. However, everything is uploaded on osf.io. For consistency reasons the authors could correct this.

Spelling comment: I encourage the authors to check the formula for the softmax activation on p.9. To my understanding a comma is missing in the brackets of the formula.

Reviewer #3: Summary:

This manuscript introduces three methods to identify the borrowed words based on monolingual wordlists. The author evaluates the performances of these methods by setting up various scenarios. Despite the methods work well in case of artificially seeded data, performances of these methods are not all satisfied.

Nevertheless, the author points out that the high proportion of borrowing and the existence of a dominant donor language in a language are beneficial to the task. Also, a more promising method for borrowing detection is recommended for future study.

Strength:

This paper applies a dataset including a large amount of languages so that we can know the characteristics of languages that are suitable for the examined methods.

It is interesting to see that the phonological and phonetic features are applied to build up the lexical models.

In terms of the Marko model and neural network model, it is also interesting to see your way to use entropy difference to classify borrowed.

The author provides useful suggestion on future works and extract meaningful information despite the unsatisfied result of the models.

Weakness:

Generally, the structure of the paper is fine, but sometimes I was surprised to see some contents that don’t belong to a section. I also saw some duplicated and redundant information. Besides, it’d be better to clearly indicate the meaning of the notations in formula. Below is the specific comments referring to specific sections.

--Abstract

You mentioned all necessary information in the abstract. However, it is not always clear and I had to spend some time to look for the information I needed. For instance, what you did and the result of this study are not clearly specified. In my opinion using phrases like “in this study, we did this…” “the result shows that…” could be helpful to catch the necessary information easier and faster before reading it.

--Introduction

You mention a lot in the introduction on how good a native speaker can identify the borrowing in a his/her own language, and it seems that this is related to your motivation to do the study. But you didn’t address this later in the rest of the paper. Hence, I'm confused about the actual motivation or the problem you try to address in the paper.

Also, it is a little wired that you include results and some discussion content at the end of the introduction, making this part like an abstract.

--Materials

It’d be easier to understand the WOLD dataset if you introduce more about the data format and show some examples. I had to search for the WOLD online to understand how the data looks like. Meanwhile, adding some phonetic transcription examples would be more interesting and easier to understand what you did.

Line 134 citation [36], there is typo for the reference of this citation in page 32/39.

--Markov Model

Page 8/39 it might be better to explain the notation. I had to spend some time to guess what they meant. The same as in Page 9

Line 202. Any reference for this statement?

Line 232 to 238 Maybe you wanna sum up the three methods, but it seems a little redundant as they have been introduced previously. Try to re-organise it and avoid duplicated content.

--Result

In general, you wrote the result for each experiment separately, which is clear and good. However, at the beginning of each section, you introduced the detail of each experiment, and the introduction of the experiments should not be part of the result section. The introduction of the experiments should be somewhere else as I only expect actual result like digits, tables, figures and relevant explanation of them in this section.

Another option is to write each experiment totally separately, meaning that you write about the introduction, result, and discussion of an experiment together. The structure of the whole paper would be clearer and more logical.

Line351 to 353, any reason you choose these three characteristics? Besides, have you considered that if they are independent from each other?

Page 17/39, Table 4, are all these correlations significant? Maybe you can also include p-value.

Page 18/39 Table 5, it is a nice table showing regression model R square value. But it’d be better to involve the information in the table when you discuss the influence of the three factors, instead of just putting it here.

--Discussion

You had to introduce each experiment again in this section, and it is redundant. As I suggest above in the result section, write about the experiment one by one, then you don’t have to constantly describe the experiments again and again.

Line 502-507 maybe you can try the frequency of these unique sounds.

6. PLOS authors have the option to publish the peer review history of their article (what does this mean?). If published, this will include your full peer review and any attached files.

Reviewer #1: No

Reviewer #2: No

Reviewer #3: **Yes: **Liqin Zhang

---

## [Author Response · Author response to Decision Letter 0]

7 Nov 2020

We have responded to editor and reviewer comments in the response to reviewers document included in the attach files step with file name RebuttalLetterPONE-D-20-27304-final.pdf

We are copying the letter contents here:

Response to Reviewers (PONE-D-20-27304)

Comments by the Editor

All three reviewers have some good suggestions that you should take into account. For Reviewer 1 it was a stumbling block that the paper seems to try to model a native speaker's ability to identify loanwords. Given that it doesn't do a good job at that the verdict was a rejection. But it seems that the paper is really about the extent to which it is possible for a computer to identify loanwords given information about the target language only. If you clearly spell out that focus and downplay the importance of discussion about what native speakers can and cannot then you might avoid some confusion.

We thank the editor for the very helpful review process and the thorough selection of reviewers. We have now tried to modify the draft consistently, especially taking the global points raised by the reviewers into account. Our modifications are specifically reflected in:

 1. No longer talking about the native speakers as our inspiration and desire for modeling, as we find that the clues we use are also used by traditional linguists who try to detect borrowings.

2. Re-structuring the presentation of the results, following mostly reviewer III’s very useful recommendations.

3. Publishing data and code openly and archiving them with Zenodo.

In addition, we carried out many minor modifications, as reflected in our detailed response below, as well as in the modified text, in which we have marked all modified text blocks in blue font.

● A rebuttal letter that responds to each point raised by the academic editor and reviewer(s). You should upload this letter as a separate file labeled 'Response to Reviewers'.

● A marked-up copy of your manuscript that highlights changes made to the original version. You should upload this as a separate file labeled 'Revised Manuscript with Track Changes'.

● An unmarked version of your revised paper without tracked changes. You should upload this as a separate file labeled 'Manuscript'.

Comments by Reviewer #1

The article is a rather mechanical application of several machine-learning algorithms, two of them severely outdated and none of them state-of-the-art, to what is essentially a non-task. The motivation for the experiments---that speakers of different languages (cited publications refer to Russian and Korean in particular) are good at identifying borrowings---is way too slim.

We understand from the reaction of the reviewer that our first draft has severely failed to make clear what the motivation behind this study was. In our updated version, we have now tried to clarify it. To summarize the changes (we will also provide detailed points below), our revision should make clear that:

1. Borrowing detection is a task that takes multiple pieces of evidence into account, but computational methods have so far not fully used all of the evidence considered by experts.

2. Our study proposes one way to operationalize language-internal evidence considered by experts, drawn from phonology and phonotactics.

3. To explore the usefulness of this evidence and our operationalization, we conduct experiments and use three different approaches, which provide varying levels of complexity, as it is standard in machine learning

 operations (one should never only test the most complex model to avoid

overfitting)

4. Our findings suggest that phonological and phonotactic clues for lexical

borrowings are not able to provide fully satisfying results in general, especially since they are mostly useful for recent borrowings, but that they are consistent with our expectation (increasing accuracy with more borrowings and with more borrowings from a unique source).

We hope that this is enough to show that we are not necessarily talking about null-results here. To further emphasize this, we explicitly point out in the conclusion, that we do not believe that research should only be based on “good results” alone, but that all results are worth being shared in order to advance science (which is also in line with the mission of PLOS).

Firstly, this only applies to very recent borrowings (Russian, just like English, is saturated with older borrowings, which are undetectable by native speakers).

This should be much clearer in our updated version now, that we are aware that phonotactic and phonological clues are strongest for recent borrowings.

Secondly, this does not provide a cross-lingual baseline against which to compare the performance of ML algorithms.

We have tried to make our point clearer by explicitly mentioning in the introduction of the new draft, that borrowing is a unique process and it is therefore difficult to generate a baseline (gold standard) for it.

Thirdly, as the authors point out themselves, this is not how borrowings are detected in the historical-comparative literature (where the principle of irregular sound correspondences is the only one of real standing; of course, this principle is rather hard to automate because one has to establish the correspondences manually to begin with).

We now emphasize in the draft, that classical historical linguists use phonology and phonotactics as one clue among many, when it comes to the detection of borrowings, and that the majority of clues are comparative in nature, that is, they require the comparison of the language that received borrowings with potential donors.

It is hard to grasp what exactly the study is trying to show. It could have been construed as an attempt to model discriminative abilities of native speakers, in which case the focus should have been on how the neural net discriminates between native and borrowed words (cf. the abundant recent literature on what BERT might know about

 syntax, etc.). Instead all the models are treated as black boxes, and the analysis boils down to identifying situations in which they perform better or worse. This may have been of interest had the proposed method been of practical use.

As mentioned before, this should be clarified in the updated draft. We hope specifically that we have done a better job now at explaining why we think that our method is of practical use.

Some possible applications are listed in the conclusion ("studies in which borrowed words or sentences need to be identified in large amounts of data"; "[work] on code switching, where multilingual language users switch between different varieties based on sociolinguistic contexts"); however, the fact that the proposed methodology can help there itself needs to be tested against appropriate baselines and competing approaches.

We are confident that the usefulness of our method has now been more properly explained in the updated version. We also explicitly mention now that — even if it turns out that our approach does not provide useful for any follow up studies or that it cannot be further improved — we think it is important to share our results, since this kind of “failed research” may help those who want to try similar approaches in the future to save some precious time, building on our efforts.

Comments by Reviewer #2

The authors introduced a new approach to automatic loanword detection in the field of computational historical linguistics (CHL). While most of the existing methods aim at identifying loanwords in multilingual wordlists, the attempt of this paper is to identify borrowings using a monolingual approach. The authors use the WOLD database, which is the only database containing loanword information along with information about the donor language and loaned status. What I really liked about the work is that the three methods build on another: the Bag of Sound model using an SVM is the simplest model, integrating only the phonology of the words without considering the order and frequency of the sounds; the Markov Model is a tri-gram model relying on the two previous sounds in the word; the recurrent neural network also relies on the phonotactics of the word, taking all previous sounds into account. The authors perform two experiments, one on artificially seeded borrowings and one on the “real” WOLD data. Since the promising results on the simulated data could not be obtained by the experiments using the WOLD data, the authors made additional experiments in order to explain the performance of the methods, which was achieved. Although, the results and the performance are not satisfying, the three introduced methods along with the lexical language models open a new perspective for further research, especially the recurrent neural network, serves as

 basis for improvements and further explorations in the field of automatic loanword detection.

Two things I really liked about the manuscript are the detailed explanation of the methods and the representation of the data, which help the reader to get clear insights in the evaluation of the different methods. The statistical evaluation are carried out and explained in detail. The authors made the effort to perform additional analyses to explain the performance of the methods in the two experiments. The statistical evaluation are carried out in a rigorous way, giving some explanations according to the performance and the results.

The process of nativization plays a crucial role in the motivation of the approach introduced in the first chapters, however it was not revive in more detail in the conclusion. The detection of loanwords using the proposed methods depends highly on the data and the adjustment status of the word in the recipient language. The methods might not identify older loanwords or loaned words from related languages, which show no clear differences in the phonology. This issue was not discussed in detail in the conclusion, but could be one of the reasons of the poor performance of the methods.

We have now largely modified the description of the approach and decided to give up the parallel to borrowing detection by native speakers. First, we found that native speakers’ knowledge may often not be as perfect as it seems (judging on our discussion among the multi-lingual team of authors and linguists and what we know about the knowledge of native speakers in our respective native tongues), and second, we found that phonological and phonotactic clues are also justified and discussed as such in the traditional linguistic literature. As a result, we no longer talk about the native speakers’ intuition, but rather emphasize the importance of testing the power of language-internal clues for borrowing detection on a larger scale.

In addition, the automatically derived IPA transcriptions of the words from the WOLD database could lead to noise in the analyses, depending on correctness of the transcription. However, since no other database is available containing loanword annotations along with the donor language and loanword status, compromises need to be made.

We have added a table for demonstrating how the data is organized in memory, and expanded the discussion on how it is organized in disk via the CLDF standard. We also expanded the discussion and the references on how the transcriptions were obtained

 (i.e., orthographic profiles) and added a note informing, as mentioned by the reviewer, that the transcription might add noise to the data.

The data is completely available online. Within the data, the additional German wordlist used for the artificially seeded borrowings is not identifiable at first sight. I would encourage the authors to provide the list in a format like csv and allocate it at first sight in the python package. Additionally, as a small notice, on p.6 the authors wrote that the German word list and the software package are available on GitHub. However, everything is uploaded on osf.io. For consistency reasons the authors could correct this.

We’ve provided an updated github/zenodo link for all the data and code. Additionally we’ve provided a reference including url to the published German wordlist.

Spelling comment: I encourage the authors to check the formula for the softmax activation on p.9. To my understanding a comma is missing in the brackets of the formula.

We’ve added the missing comma.

Comments by Reviewer #3

Summary:

This manuscript introduces three methods to identify the borrowed words based on monolingual wordlists. The author evaluates the performances of these methods by setting up various scenarios. Despite the methods work well in case of artificially seeded data, performances of these methods are not all satisfied.

Nevertheless, the author points out that the high proportion of borrowing and the existence of a dominant donor language in a language are beneficial to the task. Also, a more promising method for borrowing detection is recommended for future study.

Strength:

This paper applies a dataset including a large amount of languages so that we can know the characteristics of languages that are suitable for the examined methods.

It is interesting to see that the phonological and phonetic features are applied to build up the lexical models.

In terms of the Marko model and neural network model, it is also interesting to see your way to use entropy difference to classify borrowed.

 The author provides useful suggestion on future works and extract meaningful information despite the unsatisfied result of the models.

Weakness:

Generally, the structure of the paper is fine, but sometimes I was surprised to see some contents that don’t belong to a section. I also saw some duplicated and redundant information. Besides, it’d be better to clearly indicate the meaning of the notations in formula. Below is the specific comments referring to specific sections.

--Abstract

You mentioned all necessary information in the abstract. However, it is not always clear and I had to spend some time to look for the information I needed. For instance, what you did and the result of this study are not clearly specified. In my opinion using phrases like “in this study, we did this...” “the result shows that...” could be helpful to catch the necessary information easier and faster before reading it.

We’ve added appropriate hints to the abstract to help with reader processing of information.

--Introduction

You mention a lot in the introduction on how good a native speaker can identify the borrowing in a his/her own language, and it seems that this is related to your motivation to do the study. But you didn’t address this later in the rest of the paper. Hence, I'm confused about the actual motivation or the problem you try to address in the paper.

It is the case that perceptions of native user performance, including our own anecdotal experiences of loanword awareness, helped to inspire this study. However, consulting the linguistic literature again, and discussing the degree to which native speakers really are able to identify borrowings showed that this point of inspiration is more a potential myth than a tested fact. As a result, we’ve now removed all hints to native speakers from the new version and emphasize instead that classical linguists routinely consider phonological and phonotactic clues when it comes to the detection of borrowings. So our intention is not to model native speakers, but to provide and test a method for lexical borrowing detection that is exclusively based on mono-lingual data in a supervised framework. This, we hope, is much clearer now in the draft.

Also, it is a little weird that you include results and some discussion content at the end of the introduction, making this part like an abstract.

We reduced the text substantially while still anticipating results and discussion sections.

 --Materials

It’d be easier to understand the WOLD dataset if you introduce more about the data format and show some examples. I had to search for the WOLD online to understand how the data looks like. Meanwhile, adding some phonetic transcription examples would be more interesting and easier to understand what you did.

We’ve added a small table of transcription examples from represented languages.

Line 134 citation [36], there is typo for the reference of this citation in page 32/39.

The reference has been corrected, providing the paper’s title.

--Markov Model

Page 8/39 it might be better to explain the notation. I had to spend some time to guess what they meant. The same as in Page 9

Notation has been better explained in the text for both Markov and Neural models.

Line 202. Any reference for this statement?

We added references to Bengio’s original language modeling work using recurrent layers and the more recent, transformer language model. Both show improvements over Markov Models in language modeling and we expected similar improvements for sound segment modeling.

Line 232 to 238 Maybe you wanna sum up the three methods, but it seems a little redundant as they have been introduced previously. Try to re-organise it and avoid duplicated content.

We have reduced the redundancy and better introduced the idea that we are using dual inherited and borrowed models as part of the decision procedure.

--Result

In general, you wrote the result for each experiment separately, which is clear and good. However, at the beginning of each section, you introduced the detail of each experiment, and the introduction of the experiments should not be part of the result section. The introduction of the experiments should be somewhere else as I only expect actual result like digits, tables, figures and relevant explanation of them in this section.

We’ve reorganized the presentation of experiments and results so that each experiment and its results are treated separately from the others. This avoids the redundancy of

 multiple presentations of the same experiment. We hope this comes close to what the reviewer has suggested as the alternative option (in the next paragraph)

Another option is to write each experiment totally separately, meaning that you write about the introduction, result, and discussion of an experiment together. The structure of the whole paper would be clearer and more logical.

Thank you for your suggestion. We now follow your advice by presenting each experiment along with its results in turn, with each occurring as a subsection of the “Experiments with results” section.

Line351 to 353, any reason you choose these three characteristics? Besides, have you considered that if they are independent from each other?

We now explain the reason for choosing these characteristics in the text. The proportion of borrowed words is moderately negatively correlated with the proportion of unique native sound segments, otherwise the characteristics are independent.

Page 17/39, Table 4, are all these correlations significant? Maybe you can also include p-value.

We wanted to emphasize the strength of the relationships between phonological characteristics and borrowing prediction performance. So we noted in the table of correlations that all magnitudes >= 0.33 are significant at p < 0.05. We hope this is satisfying enough for the reviewer.

Page 18/39 Table 5, it is a nice table showing regression model R square value. But it’d be better to involve the information in the table when you discuss the influence of the three factors, instead of just putting it here.

The table information has now been explicitly referenced in the text of the results section.

--Discussion

You had to introduce each experiment again in this section, and it is redundant. As I suggest above in the result section, write about the experiment one by one, then you don’t have to constantly describe the experiments again and again.

We decided to keep the discussion section separate from the experiments. With the changes made, following the reviewer’s suggestion above regarding the organization of the text by experiment (and more careful editing), we hope we succeeded in reducing the redundancy of the presentation.

Line 502-507 maybe you can try the frequency of these unique sounds.

Thanks for this hint. The text now (hopefully) interprets the results better (especially of the univariate and multiple regression analyses), by including the impact of exclusively borrowed word sounds.

---

## [Editor Report · Decision Letter 1]

9 Nov 2020

Using lexical language models to detect borrowings in monolingual wordlists

PONE-D-20-27304R1

Dear Dr. Tresoldi,

We’re pleased to inform you that your manuscript has been judged scientifically suitable for publication and will be formally accepted for publication once it meets all outstanding technical requirements.

Kind regards,

Søren Wichmann, PhD

Academic Editor

PLOS ONE

Additional Editor Comments (optional):

The revisions look fine, and should be satisfactory for the reviewers, so a second round of reviewing is not necessary. I noted a couple of typos/stylistic issues that you could fix:

depends from the initial contact situation -> depends on the initial contact situation

and (b) the more borrowings go back -> and (b) when more borrowings go back [or something like that]
---

## [Editor Report · Acceptance letter]

16 Nov 2020

PONE-D-20-27304R1 

Using lexical language models to detect borrowings in monolingual wordlists 

Dear Dr. Tresoldi:

I'm pleased to inform you that your manuscript has been deemed suitable for publication in PLOS ONE. Congratulations! Your manuscript is now with our production department. 

Kind regards, 

on behalf of

Dr. Søren Wichmann 

Academic Editor

PLOS ONE